# Mussel Mass Mortality and the Microbiome: Evidence for Shifts in the Bacterial Microbiome of a Declining Freshwater Bivalve

**DOI:** 10.3390/microorganisms9091976

**Published:** 2021-09-17

**Authors:** Jordan C. Richard, Lewis J. Campbell, Eric M. Leis, Rose E. Agbalog, Chris D. Dunn, Diane L. Waller, Susan Knowles, Joel G. Putnam, Tony L. Goldberg

**Affiliations:** 1Department of Pathobiological Sciences and Freshwater & Marine Sciences Program, University of Wisconsin-Madison, Madison, WI 53711, USA; jcrichard2@wisc.edu (J.C.R.); lewis.campbell@wisc.edu (L.J.C.); cddunn2@wisc.edu (C.D.D.); 2Southwestern Virginia Field Office, US Fish and Wildlife Service, Abingdon, VA 24210, USA; rose_agbalog@fws.gov; 3La Crosse Fish Health Center-Midwest Fisheries Center, US Fish and Wildlife Service, Onalaska, WI 54650, USA; eric_leis@fws.gov; 4Upper Midwest Environmental Sciences Center, U.S. Geological Survey, La Crosse, WI 54603, USA; dwaller@usgs.gov (D.L.W.); jgputnam@usgs.gov (J.G.P.); 5National Wildlife Health Center, U.S. Geological Survey, Madison, WI 53711, USA; sknowles@usgs.gov

**Keywords:** bivalve, 16S rRNA, microbiome, freshwater mussel, mass mortality, die-off, unionid, yokenella, aeromonas

## Abstract

Freshwater mussels (Unionida) are suffering mass mortality events worldwide, but the causes remain enigmatic. Here, we describe an analysis of bacterial loads, community structure, and inferred metabolic pathways in the hemolymph of pheasantshells (*Actinonaias pectorosa*) from the Clinch River, USA, during a multi-year mass mortality event. Bacterial loads were approximately 2 logs higher in moribund mussels (cases) than in apparently healthy mussels (controls). Bacterial communities also differed between cases and controls, with fewer sequence variants (SVs) and higher relative abundances of the proteobacteria *Yokenella regensburgei* and *Aeromonas salmonicida* in cases than in controls. Inferred bacterial metabolic pathways demonstrated a predominance of degradation, utilization, and assimilation pathways in cases and a predominance of biosynthesis pathways in controls. Only two SVs correlated with Clinch densovirus 1, a virus previously shown to be strongly associated with mortality in this system: Deinococcota and Actinobacteriota, which were associated with densovirus-positive and densovirus-negative mussels, respectively. Overall, our results suggest that bacterial invasion and shifts in the bacterial microbiome during unionid mass mortality events may result from primary insults such as viral infection or environmental stressors. If so, bacterial communities in mussel hemolymph may be sensitive, if generalized, indicators of declining mussel health.

## 1. Introduction

North American freshwater mussels (family Unionidae) are a diverse (~300 species) and highly imperiled group that contribute important ecosystem services to freshwater habitats. Approximately two thirds of these species are threatened, endangered, or vulnerable, while 10% are already extinct [1], making them the most imperiled North American faunal group [2,3]. As these species decline, there is a corresponding loss of valuable ecosystem services provided by mussels, which include increased physical habitat complexity [4], filter-feeding and removal of suspended particulate matter [5], nutrient deposition coupling benthic to pelagic habitats [6], and food web enhancement [7]. Many factors have been linked to freshwater mussel population declines [8], including habitat destruction and loss due to river impoundment and pollution, over-harvest for commercial use [9], and competition from nonnative species (particularly the Asian clam *Corbicula fluminea*, zebra mussel *Dreissena polymorpha*, and quagga mussel *D. rostriformis bugensis*) [10]. Although they were historically a food resource for Native Americans [11], North American Unionids are not typically consumed by people today. However, many Unionid population declines and mass mortality events (MMEs) are enigmatic in that they have occurred without obvious causes [12,13].

In other taxa experiencing enigmatic declines or MMEs, investigations have demonstrated associations between increased mortality and infectious diseases, e.g., amphibian chytridiomycosis [14,15], white-nose syndrome in bats [16], and devil facial tumor disease in Tasmanian devils [17]. In marine bivalves, MMEs are often caused by infectious agents, with prominent examples including viruses (*Ostreid herpesvirus 1* (OsHV-1)) [18], bacteria (particularly members of the genus *Vibrio*) [19], and protists (*Haplosporidium nelsoni*, *Perkinsus marinus*) [20,21]. Like many Unionids, the marine bivalve *Pinna nobilis* experienced widespread anthropogenically induced population declines beginning in 1980 [22], followed by a series of mass mortality events beginning in 2016 [23]. These mortalities have been putatively attributed to pathogens including *Haplosporidium pinnae* [24], *Mycobacterium* sp. [25], and opportunistic *Vibrio* spp. [26]. Infectious disease outbreaks in marine bivalves are most often driven by interactions among host factors, abiotic environmental conditions, and pathogens [27,28].

Freshwater bivalves are also suffering global declines and MMEs, but data on infectious causes are comparatively sparse [29,30]. Previous investigations of mussel MMEs in the Tennessee River in 1985 and 2008 demonstrated differential abundances of an unidentified bacterium with significantly elevated loads in moribund mussels [31] and significantly elevated total bacterial loads from moribund mussels in a multi-year epidemiological study from 2006–2008 [32]. Recent work for both marine and freshwater bivalves has characterized microbiomes using 16S rRNA metabarcoding to explore relationships among bivalves, their associated bacteria, and the environment. Freshwater mussel gut microbiomes differ in wild and captive habitats [33,34], are functionally stable [35], are determined by host species, and are selectively retained from filtered seston [36].

The Clinch River in Southwestern Virginia and Northeastern Tennessee, USA, holds 46 extant species of freshwater mussels (of which 20 are federally listed as endangered), making it among the most biodiverse and ecologically important rivers in North America. An ongoing MME in the Clinch River has reduced the population of one of the most abundant mussel species, the pheasantshell *(Actinonaias pectorosa*), by >80% across multiple sites throughout the lower river [37]. This system is part of an ongoing, multi-year effort to understand the causes of MMEs [38]. Our previous investigations demonstrated a strong association between health status of pheasantshells (dying versus apparently healthy) in the Clinch River and a novel virus, Clinch densovirus 1, with viral loads 2.7 (log10) times higher in cases than in controls and cases 11.2 times more likely to be positive for the virus than controls [39]. Additionally, the bacterium *Yokenella regensburgei* has been identified in the same context [40]. These findings suggest that viruses may interact with bacteria during MMEs, which has been shown for marine bivalves using culture-independent methods for bacterial community assessment but not to our knowledge for freshwater bivalves. The purpose of this study was to (1) characterize bacterial communities in the hemolymph of pheasantshells from the Clinch River using an epidemiologic case-control study design, (2) identify bacterial taxa that warrant further investigation as potentially contributing to pheasantshell mortality, and (3) investigate associations between microbial community composition and infection with Clinch densovirus 1.

## 2. Materials and Methods

### 2.1. Experimental Design and Field Sampling

Field sampling methods were previously described [39]. Briefly, we collected hemolymph samples from the anterior adductor muscle of pheasantshell from the Clinch River during periods of observed mass mortality in 2017 and 2018. We chose to sample hemolymph because it can be collected non-lethally and is considered a preferred sample for unionid health assessments [41]. Samples were collected using an epidemiological case-control design to compare moribund mussels (animals observed lying on the substrate surface, gaping, and minimally responsive to tactile stimulation, Appendix A) from four sites at which MME was observed to apparently healthy mussels (buried in the substrate, siphoning normally, and strongly responsive to tactile stimulation) from these same sites and from two upstream sites at which no MME was observed (Figure 1, Appendix A). Hemolymph samples were frozen on dry ice immediately after collection in the field and transported for long-term storage at −80 °C until laboratory processing.

### 2.2. DNA Extraction, PCR, and Sequencing

Laboratory and analytical methods closely follow those of Dunn et al. (2020) [42], with minor adaptations for mussel hemolymph. Bacteria in hemolymph samples were pelleted by centrifugation at 10,000× *g* for 10 min. Following the removal of supernatants, we extracted nucleic acids from pellets using the Qiagen AllPrep PowerViral DNA/RNA Kit (Qiagen, Hilden, Germany), following the manufacturer’s protocol and eluting a 50 µL volume. Next, we conducted polymerase chain reaction (PCR) amplification targeting the V4 region of the 16S ribosomal RNA gene using protocols developed by the Earth Microbiome Project [43]. Each PCR contained 2 µL DNA template, 9.5 µL of Qiagen Nuclease free water, 12.5 µL of Qiagen HotStarTaq master mix, 0.5 µL of 10 mM forward primer 515f (5′-TCGTCGGCAGCGTCAGATGTGTATAAGAGACAGGTGYCAGCMGCCGCGGTAA3′), and 0.5 µL of 10 mM reverse primer 806Rb (5′-GTCTCGTGGGCTCGGAGATGTGTATAAGAGACAGGGACTACNVGGGTWTCTAAT-3′) [44,45] in a total volume of 25 µL. Cycling conditions consisted of an initial denaturation step of 15 min at 95 °C, followed by 35 cycles of 30 s at 94 °C, 30 s at 58 °C, and 1 min at 72 °C, and a final extension of 10 min at 72 °C. PCR products were prepared for sequencing in multiplex using the Nextera XT Index v2 Kit (Illumina, San Diego, CA, USA), following manufacturers standard instructions. We visualized samples after barcoding via electrophoresis on a 2% agarose gel. To remove excess index and amplification primers from our PCR products, we excised amplicons from gels under ultraviolet light using a biopsy punch and extracted them using the Zymoclean gel DNA recovery kit (Zymo Research, Irvine, CA, USA). The DNA concentration of each sample library was determined using a Qubit fluorometer (Invitrogen, Waltham, MA, USA), and sample libraries were pooled at equal molarity. We then performed a magnetic bead clean-up, using a 0.7-ratio of AMPure beads (Beckman Colter, Indianapolis, IN, USA) to pooled library volume. We sequenced libraries on an Illumina MiSeq instrument (Illumina, San Diego, CA, USA), using v3 paired-end 2 × 300 cycle chemistry. We included negative controls throughout our extraction and amplification procedures and used the resulting data to subtract potential contaminants in silico using the R package decontam [46]. Negative controls comprised DNA extractions with no input hemolymph and PCR amplifications using 2 µL of sterile molecular grade water in place of DNA template.

### 2.3. Bacterial DNA Quantification

We used the Femto Bacterial DNA Quantification Kit (Zymo Research, Irvine, CA, USA) to quantify the extracted bacterial DNA in hemolymph samples from each individual mussel. Each Quantitative PCR (qPCR) sample comprised a 20 μL volume containing 18 μL of Femto Bacterial qPCR premix and 2 μL of either extracted sample DNA, bacterial DNA standard, or no template control. Thermocycling conditions were as follows: initial incubation of 95 °C for 10 min, followed by 40 cycles of 95 °C for 30 s, 50 °C for 30 s, 72 °C for 1 min, and a final extension at 72 °C for 7 min. We ran qPCRs in triplicate and used the average calculated starting concentration across all three runs as a measure of bacterial load for each sample.

### 2.4. Bacterial Community Assessment

We quality-screened and trimmed sequence reads using the R [47] package DADA2 [48]. Raw sequence file (FASTQ) reads were trimmed at the first base with a quality score ≤ 2. We truncated forward and reverse reads at 240 and 160 bases, respectively, to account for differential rates of per-base quality degradation between forward and reverse reads. We also removed reads with non-assigned bases (N) and reads mapping to the PhiX sequencing standard. We used DADA2 to detect sequence variants (SVs), merge paired reads into single consensus reads, and remove chimeric sequences. We used the R package decontam to identify and remove SVs associated with negative control samples. We evaluated decontam results from both the prevalence and concentration methods and chose the more conservative of the two (i.e., the method that identified more SVs as potential contaminants). SVs removed at this step were identified from the genera *Escherichia*-*Shigella*, *Staphylococcus*, *Corynebacterium*, *Acinetobacter* (two SVs), *Streptococcus*, and *Anaerococcus* (two SVs), most of which have been previously identified as common contaminant taxa in extraction blank controls and no-template controls [49]. Following quality filtering, merging of overlapping reads, and removal of blank and chimeric sequences, the dataset contained 7,239,230 reads and 2490 SVs.

We assigned taxonomy to SVs at the genus (and species, where possible) level using the SILVA ribosomal RNA database [50]. We then removed all SVs assigned to eukaryotic organisms and singletons (SVs that appeared only once in the dataset). We used an abundance-based filter to remove SVs with <100 reads across all samples in the dataset to avoid biases associated with extremely rare SVs. Following these steps, the resulting dataset contained 5,489,832 reads from 231 SVs in 59 mussel hemolymph samples.

Prior to analyses using the R package phyloseq [51], we omitted mussel samples with <1000 reads (*n* = 3) to ensure sufficient minimum reads per sample for characterizing bacterial communities. After removing these samples and any resulting singleton or zero-read SVs, the final dataset for analysis contained 5,446,945 reads from 204 SVs in 56 mussel samples.

### 2.5. Statistical Analysis

All comparative analyses were based on differences in two metrics: clinical status (case versus control) and densovirus status (animals classified as positive versus negative for Clinch densovirus 1). Densovirus status for each individual was determined using metagenomic methods for virus identification, as described previously in [39]. Three individuals in this study were not sequenced in the previous virology study and were omitted from tests based on densovirus status, resulting in slightly different analysis groups. We used a set of three complementary metrics to assess alpha diversity in the dataset: a basic raw count of SVs per sample (SV richness), Shannon index, and the Inverse Simpson index. Prior to alpha diversity assessment, we used the rarefy function in phyloseq to randomly subsample (rarefy) individual libraries from the input dataset down to the smallest set of reads in a sample (1289 reads) to account for biases introduced by differences in individual library sizes. We used the three metrics of alpha diversity to test for differences based on clinical and densovirus status. For each test, we assessed the normality of the response variable distribution with Shapiro–Wilk tests and the homogeneity of group variances with Bartlett’s tests. We used Wilcoxon rank sum exact tests for comparison of groups using Inverse Simpson and Observed SV indices, which were non-normally distributed but had equal variances between test groups. Shannon Index tests fit normality assumptions for response variable distribution, but group variances were equal for clinical status and unequal for densovirus status groups, resulting in our use of Welch’s two sample *t*-test and a two-sample *t*-test assuming equal variances, respectively.

To evaluate Beta diversity, we used the vegan package in R [52] to create a non-metric multidimensional scaling (NMDS) ordinations based on Bray–Curtis dissimilarity distances calculated between individual sample bacterial community pairs. We iteratively constructed ordination models using between 1 and 10 axes and examined stress plots to determine the optimum number of axes to use. We used the R package ggplot to visualize ordinations and examine trends based on grouping variables of interest, including sample date, sample site, clinical status, reproductive status (gravid or not gravid), densovirus status, and shell length. We then tested for differences in community structure based on these variables using permutational analysis of variance tests (PERMANOVA) using the adonis function in vegan with 999 permutations. We constructed stacked bar graphs of abundances of the top 20 genera and plotted them to visualize common taxa and differences among sampled individuals.

To detect differentially abundant SVs (indicator SVs) between groups based on clinical and densovirus status, we used linear discriminant analysis effect size (LEfSe; [53]). LEfSe incorporates a two-stage procedure to first determine which SVs are differentially abundant between groups and then assess which of those SVs are consistently present among individuals of a given group. Prior to analysis with LEfSe, we converted raw SV data into relative abundances for each individual. We also conducted LEfSe analysis with the data summarized at the phylum level, using the glom function in Phyloseq, to explore higher-order differences between sample groups. All LEfSe analyses were conducted using an LDA threshold of 2.0 and significance cutoff of <0.05. Following identification of indicator SVs, we used BLAST [54] to check highly prevalent SVs and indicator SVs against the GenBank 16S rRNA database to assess closely related matches.

We estimated potential bacterial community functionality based on marker gene data using PICRUST2 [55] with default parameters. Input files were generated using the biomformat package [56] in R, the PICRUST2 algorithm was implemented in python, and output files were summarized based on hierarchical classes in the MetaCyc database [57] using the command “--custom_map_table”. We analyzed output data tables describing MetaCyc pathways at the highest (superpathway), second-highest, and individual pathway levels to assess both individual pathways and more general differences in inferred metabolic functions between comparison groups. Data tables for each level were converted to percent abundance for each individual and used as inputs for LEfSe analysis with LDA threshold of 2.0 and significance cutoff of <0.05. A weighted Nearest Sequenced Taxon Index (NSTI) score for each sample was used to assess the accuracy of the inferred pathways, as the accuracy depends primarily on the availability of closely related reference genomes [58].

Finally, we calculated basic measures of prevalence and percent composition (i.e., percentage of reads for an individual attributed to each SV) for the most common SVs in the dataset and indicator SVs. These measures were used to summarize representation and potential trends in prevalence or bacterial community dominance based on clinical and densovirus status. Bacterial load data measured by qPCR were assessed by comparing raw and log_10_-transformed values for calculated starting bacterial concentrations based on clinical and densovirus status.

## 3. Results

### 3.1. qPCR Data

Bacterial loads were approximately 400 times higher in case versus control mussels (Figure 2), but there were no difference between groups based on densovirus status. Cases averaged 9.55 × 10^−3^ ng of bacterial DNA per µL of hemolymph, while controls averaged 2.31 × 10^−5^ ng/µL. Several case mussels (*n* = 3) were extreme outliers and were excluded from statistical tests for differences between groups. The most extreme of these outliers had a bacterial load > 4300x higher than the average of all other samples. Excluding the 3 high outliers from the cases, case mussels still had average bacterial loads 41.2× higher than controls. Bacterial loads did not differ based on densovirus status.

### 3.2. Bacterial Community Composition

Pheasantshell hemolymph bacterial communities contained a limited number of common taxa and a large number of rare taxa (Appendix A). Of the 204 SVs included in the final analysis, only 5.4% (*n* = 11) were found in more than 20% of individuals sampled. The top 20 SVs comprised 85.3% of all reads sequenced. The top three phyla accounted for 91.8% of all sequence reads, with nine additional phyla collectively comprising the remainder of the reads (Figure 3). The top phyla included Proteobacteria (63.9%), Firmicutes (19.3%), and Actinobacteriota (8.7%). Proteobacteria included two classes: Alphaproteobacteria and Gammaproteobacteria (11.8% and 52.1% of all reads, respectively). Firmicutes largely consisted of two classes, Bacilli and Clostridia (2.8% and 16.4% of all reads, respectively), while Actinobacteriota mostly consisted of Actinobacteria (7.9% of all reads). LEfSe analysis at the phylum level indicated significant associations between control mussels and phyla Actinobacteriota and Cyanobacteria, while case mussels were associated with bacteria from phyla Proteobacteria, Verrucomicrobiota, and Deinococcota (Figure 4). Mussels positive for Clinch densovirus 1 were associated with Deinococcota.

No SVs were found across all samples. SV3, identified as *Clostridium sensu stricto 1 chauvoei* (Phylum Firmicutes), was by far the most prevalent. It had equal prevalence in cases and controls (60.9% and 57.6%, respectively) and on average constituted 23.0% (cases) and 32.1% (controls) of the reads for a given individual when present. SV7, identified as *Corynebacterium ureicelerivorans* (Phylum Actinobacteriota), was the next most abundant SV, and was present in 35.7% of samples. LEfSe analysis demonstrated significant associations between 12 SVs and either case (*n* = 5) or control (*n* = 7) mussels (Appendix A). SVs associated with cases were members of phyla Bacteroidota (*n* = 1), Deinococcota (*n* = 1), and Proteobacteria (*n* = 3), and SVs associated with controls were members of phyla Actinobacteriota (*n* = 2), Cyanobacteria (*n* = 1), and Proteobacteria (*n* = 4). Of the most abundant SVs (i.e., those present in >20% of individuals sampled), four SVs were not significantly associated with either case or control mussels. These SVs included three members of phylum Firmicutes and one of phylum Actinobacteriota and were approximately equally distributed in cases in controls with respect to prevalence and relative abundance. Five SVs were associated with densovirus status, though three of these were present in ≤7 individuals. Of the two higher-prevalence SVs associated with densovirus status, SV41 (Deinococcota) was associated with densovirus-positive mussels while SV7 (Actinobacteriota) was associated with densovirus-negative mussels. These were the only two SVs associated with both densovirus status and clinical status.

Alpha diversity varied little between cases and controls. Shannon Index and Inverse Simpson did not differ significantly based on case/control or densovirus status. Mean SV richness (±SD) was significantly lower in case mussels (9.0 ± 7.1) (*t*-test, *t* = −1.67, df = 36, *p* = 0.02) than in control mussels (11.8 ± 4.9) but did not differ significantly between mussels based on densovirus status (Wilcoxon rank sum exact test, w = 376, *p* = 0.51). Visual appraisal of abundance patterns at the phylum level (Figure 3) and the top 20 genera (Appendix A) showed elevated Proteobacteria relative abundances in cases compared to controls, which appears to be driven by higher abundances attributed to *Salmonella* and *Aeromonas*. At the genus level, both cases and controls had large proportions of their bacterial community composition (mean = 25.0%) from taxa outside the top 20 most abundant genera. NMDS ordination with three axes resulted in a stress value of 0.18. Additional axes reduced overall stress values, but only marginally. NMDS plots revealed clear visual separation of cases and controls along NMDS axis 1, but not on axes 2 or 3 (Figure 5). Visualization of NMDS plots did not reveal clear patterns of separation for any other variables assessed (sample site, sample date, and densovirus status). When the complete bacterial community was considered, PERMANOVA showed statistically significant differences in community structure based on clinical status (F = 1.51, *p* = 0.032) and sampling event (F = 1.78, *p* = 0.001) and marginal differences based on shell length (F = 1.42, *p* = 0.048) and site (F = 1.19, *p* = 0.058). Densovirus status was not a significant predictor of bacterial community structure when modeled as either a binary variable (positive or negative) or quantitatively using metagenomic-based viral loads from [39].

BLAST results for indicator SVs and highly prevalent SVs generally matched those assigned by the SILVA database using the DADA2 pipeline (Appendix A). The most prominent differences revealed by the BLAST searches were for the top two SVs associated with cases. SV4 was classified using the DADA2 pipeline as *Aeromonas salmonicida* (100% sequence identity), but BLAST results revealed numerous *Aeromonas* species with 100% matches to the query SV sequence. SV2 was initially classified as *Salmonella enterica* (100% sequence identity) using the DADA2 pipeline, but the top BLAST result was *Yokenella regensburgei* (100% sequence identity), although the ordering is likely be arbitrary, as BLAST scores were identical between three *Y. regensburgei* and seven *S. enterica* reference entries. Several other SVs shared 100% identity with multiple congeneric species, but only SV4 included assigned taxonomic results that matched equally well to different genera. SV7, which showed strong associations between the control and densovirus-negative groups, was 100% identical to several *Corynebacterium* species. SV41, which was significantly associated with both the case and densovirus-positive groups, was 100% identical to *Thermus scotoductus*.

### 3.3. Inferred Functional Profiles

Functional profiles generated from 231 input SVs identified 384 pathways from the MetaCyc database, which were summarized into six high-level groups and 52 s-level groups. The mean NSTI Score was 0.05 (SD ± 0.10) and did not differ significantly based on clinical status (*t*-test, *t* = −0.31, df = 43, *p* = 0.37) or densovirus status (*t*-test, *t* = −0.83, df = 34, *p* = 0.39). These values indicate reasonable predictive accuracy, as they are close to values known for well-covered microbiome samples, which are generally below 0.05 [58]. At the highest level, bacterial communities in cases were associated with pathways of degradation, utilization, and assimilation, while those in controls were associated with biosynthesis pathways. Densovirus-negative mussels were not significantly associated with any top-level pathway groups. Case mussels were associated with second-level pathways of fatty acid and lipid biosynthesis, degradation of amino acids and d-glucarate/d-galactarate, and superpathways involving histidine and aspartate (Figure 6). The association of control mussel bacterial communities with top-level biosynthesis pathways was driven most strongly by biosynthesis of nucleosides and nucleotides, secondary metabolites, and aromatic compounds. Other pathways associated with control mussel bacterial communities included respiration, glycolysis, and degradation of alcohols and aldehydes. Bacterial communities of densovirus-positive mussels were associated with the histidine superpathway, whereas those of negative mussels were associated with the cell structure and biosynthesis pathway. These first- and second-level pathways were driven by associations at the individual pathway level: 149 individual pathways showed significant differences based on clinical status, 94 pathways were significantly associated with controls, and 55 pathways were associated with cases. Nineteen pathways were significantly different based on densovirus status, with more associations aligning with the densovirus negative (*n* = 14) than positive (*n* = 5) group.

## 4. Discussion

Our analyses reveal differences in bacterial community abundance, structure, and function based on the clinical status of pheasantshells in the Clinch River (case or control) but not based on infection status with Clinch densovirus 1. We observed significantly higher bacterial loads in cases than in controls. Overall community structure differed between cases and controls, with fewer SVs observed in cases. Cases were associated with higher relative abundances of several members of Proteobacteria, including two particularly prominent SVs (SV2 and SV4) identified as *Salmonella*
*enterica* and *Aeromonas salmonicida*, respectively. Patterns of inferred bacterial community function also differed between cases and controls, with pathways of degradation, utilization, and assimilation dominating cases and pathways of biosynthesis dominating controls but fewer inferred functional differences observed between densovirus-negative and -positive samples.

Our observation of significantly higher bacterial loads in moribund mussels than in apparently healthy controls parallels that of previous studies of bacteria associated with unionid mortality events [31,32,59]. The relatively low bacterial concentrations in control mussels—which were often near the minimum thresholds for qPCR quantification—suggest that in healthy mussels, hemolymph is relatively sterile. Furthermore, bacteria found in the hemolymph of healthy mussels are similar to those found in the freshwater mussel gut, suggesting a low level of connectivity between the mussel gut and hemolymph compartments [35,36]. Conversely, patterns exhibited by case mussels are suggestive of bacterial invasion into the hemolymph, perhaps indicating the breakdown of normal cellular barriers to bacterial entry. Anecdotally, we have noticed that hemolymph collected from healthy mussels generally appears clear. However, some individuals yield hemolymph that has a “cloudy” appearance, and this occurs most often in moribund individuals. One such individual included in this study had a bacterial load > 4300x higher than the average of all other samples in the study, with a bacterial community dominated by *Aeromonas* spp. (51.5% of reads). It remains unclear whether such elevated bacterial loads are the result of a primary bacterial infection or if they occur as a sequela of some other insult. It is also possible that the patterns we observed in hemolymph originate from other tissues. Mucosal barriers (such as those present on the gills, mantle, and digestive gland) are primary defenses against bacterial invasion in filter-feeding organisms and are the initiation sites of almost all infectious diseases affecting bivalve molluscs [60,61]. Microbial communities in hemolymph could indirectly reflect microbial communities of such organs and barriers. If so, examining additional tissues could yield patterns stronger than we have documented for hemolymph.

The Proteobacteria dominated bacterial communities, comprising 7 of the 12 indicator SVs separating cases from controls. Gammaproteobacteria on average accounted for 32.7% of reads in all mussel bacterial communities (50.7% in cases, 20.1% in controls) and reads from 5 of 21 cases comprised >97.5% Gammaproteobacteria. Alphaproteobacteria were more evenly distributed, comprising an average of 10.3% and 9.9% of reads in cases and controls, respectively. A previous study found strong shifts in unionid gut microbiome composition when comparing wild Ohio pigtoe (*Pleurobema cordatum*) to individuals held in mesocosms under conditions known to induce wasting syndromes and mortality, with shifts to dominance by Proteobacteria in the latter [34]. Proteobacteria were also the most abundant taxa in the gut bacterial communities of the marine bivalve *P. nobilis* during a mass mortality event in the Mediterranean Sea [26]. Based on both prevalence and percent composition, SV2 (*Yokenella/Salmonella*) and SV4 (*Aeromonas* spp.) appear to be prominent indicators of morbidity and mortality during the ongoing Clinch River mass mortality event. Other SVs associated with cases by LEfSe analysis were present at much lower prevalence and/or abundance.

A multi-year case-control study of ebonyshell (*Fusconaia ebena*) MMEs in the Tennessee River found bacterial counts approximately 100x higher in moribund mussels than in controls, with a predominance of *Aeromonas* spp. (including *A. sobria*, *A. schubertii*, and *A. veronii*) and *Hafnia alvei* [32]. In our study, case mussels frequently had >95% of reads attributed to a single SV. SV2 was classified as indistinguishable between either *Salmonella enterica* or *Yokenella regensburgei* based on marker gene sequencing. A previous study by our group [40] cultured bacteria from freshwater mussel hemolymph. Of nine mussels that were examined both in that study and the current study, seven yielded *Y. regensburgei* by culture and also had SV2 present as 58.9% of reads on average. The remaining two individuals had no reads from SV2 nor any *Y. regensburgei* in cultures. *Salmonella enterica* was not cultured from any samples in the previous study. These data suggest that SV2 corresponds to *Y. regensburei. Y. regensburgei* was also identified from one sample of moribund ebonyshell from the Tennessee River during investigations of a similar MME [32]. *Y. regensburgei* is phenotypically similar to *H. alvei* and biochemically similar to both *H. alvei* and *S. enterica*, and previous case reports have reported misidentification of *Y. regensburgei* as either *Salmonella* spp. or *H. alvei* [62], leading to speculation that human infections by *Y. regensburgei* may be underestimated [63]. From the nine-sample overlap between the previous and current studies, we identified SV4 (*Aeromanas* spp.) from four samples, while the corresponding cultured samples identified *Aeromonas* spp. from three of these four.

Our results for apparently healthy individuals (controls) are similar to previous work characterizing freshwater mussel gut microbiomes. *Clostridium* appears to be a core constituent of freshwater mussel bacterial communities, being the most abundant genus in gut samples from six unionid species [36]. In our study, cyanobacteria were associated with controls, which may be related to seasonality, as significant SVs assigned to cyanobacteria were identified almost exclusively in the earlier sampling months (August and September). Given that observed morbidity and mortality began in September and peaked in October (no moribund cases were observed in August), the association between cyanobacteria and healthy controls may be an artifact of the seasonality of MMEs. However, it is noteworthy that cyanobacteria were absent from all moribund mussels collected in September. Seasonal shifts in temperature, flow, and nutrients related to the seasonal summer to fall turnover may be direct or indirect drivers of both cyanobacteria abundance and mussel mortality. Although the role of cyanobacteria, if any, in mussel mortality remains unclear, our data suggest that riverine systems may offer a useful comparison to lacustrine systems, where cyanobacteria have been studied extensively for their role in harmful algal blooms [64].

Our results concerning inferred bacterial metabolic pathways demonstrate a predominance of degradation, utilization, and assimilation pathways in cases and a predominance of biosynthesis pathways in controls. These results suggest that bacterial communities may shift from anabolic to catabolic as mussels become diseased. For black-lipped pearl oysters (*Pinctada margaritifera*), similar analyses to ours found different inferred bacterial metabolic pathways in different tissues [65], suggesting that hemolymph may not be representative of other tissues. We also caution that inferred pathways are only indirect proxies for actual metabolic function.

Overall, we found little relationship between hemolymph bacterial communities and Clinch densovirus 1, although significant associations between SV41 (*Thermus scotodoctus*) and morbidity and densovirus infection warrant further investigation. Previous studies have identified members of the *Deinococcus*-*Thermus* group in freshwater mussel gut microbiomes [35], but their significance remains unknown. For Pacific oysters (*Crassostrea gigas*), recent work has demonstrated that mortality is the result of sequential infection by OsHV-1 µVar, which causes immune suppression, followed by a proliferation of bacterial pathogens, including *Vibrio* spp. [66]. Previous studies have repeatedly characterized individual associations between oyster MMEs and OsHV-1 µVar, *Vibrio* bacteria, population genetics, and environmental factors, while more recent work has explored the importance of considering the multi-factorial nature of such events [27,28]. In the case of Clinch River pheasantshells, Clinch densovirus 1 and/or other primary insults may initiate a series of events leading to secondary bacterial invasion and ultimately mortality. Recent evaluations of water quality and its effects on unionid health in the Clinch River have not suggested any clear environmental or anthropogenic stressors that could explain the observed mortality [67]. However, such stressors likely contribute as co-factors, such that future studies should evaluate their interaction with biotic factors in the ongoing mortality event. Controlled laboratory studies (e.g., simulated environmental stressors, experimental infections with Clinch densovirus 1, *Yokenella regensburgei*, and pathogenic *Aeromonas* species) will likely be necessary to elucidate pathogenic mechanisms.

Our study has certain inherent limitations. Sampling occurred during a 3-month period, which precludes our ability to describe broader seasonal trends in bacterial community composition. Ambiguous taxonomic identification of key SVs related to mortality demonstrates the need for methods other than 16S rRNA sequencing. Similarly, studies of metabolic pathways, metabolites, and other direct measures of bacterial community function would improve our understanding of underlying pathogenic mechanisms. Future work should also consider longitudinal sampling and the deployment of sentinel individuals into natural habitats.

Despite these caveats, our results suggest a scenario in which a primary insult such as viral infection or an environmental stressor leads to a weakened physiological state and increased susceptibility to bacterial invasion of the otherwise-sterile hemolymph. If so, increased bacterial loads and shifts in bacterial community structure and function may be sensitive indicators of declining mussel health and thus useful for population monitoring, even when the primary cause of MMEs remains unclear.

## Figures and Tables

**Figure 1 microorganisms-09-01976-f001:**
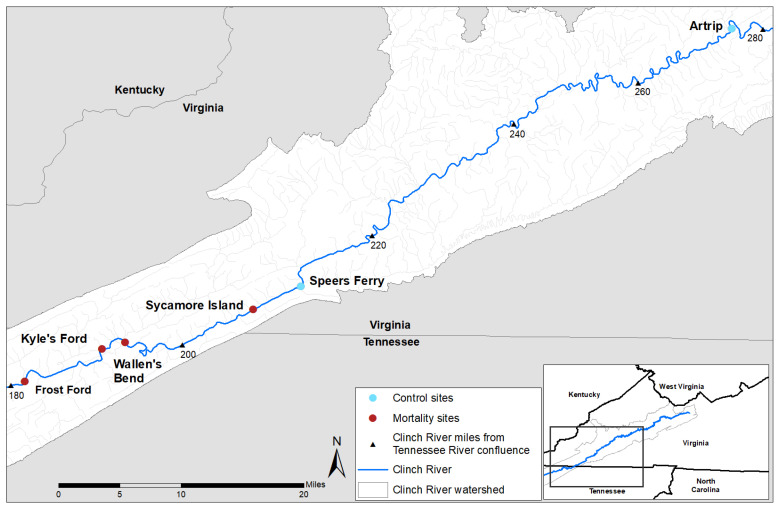
Map of sampling locations on the Clinch River, Virginia and Tennessee, USA.

**Figure 2 microorganisms-09-01976-f002:**
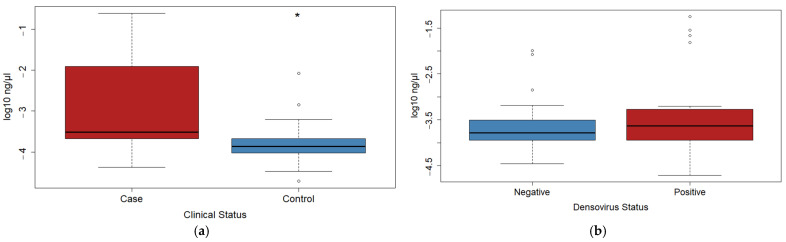
Boxplots of log_10_ transformed bacterial loads (ng/µL of hemolymph), as measured by qPCR with the Zymo Femto Bacterial DNA Quantification Kit. Three high outliers were excluded from cases in this figure and corresponding statistical test. (**a**) Bacterial loads were significantly higher in cases than in controls (Wilcoxon rank sum exact test, *p* < 0.001). Asterisk indicates statistically significant difference; (**b**) bacterial loads were not significantly different between the densovirus-negative and densovirus-positive groups (Wilcoxon rank sum exact test, *p* = 0.434).

**Figure 3 microorganisms-09-01976-f003:**
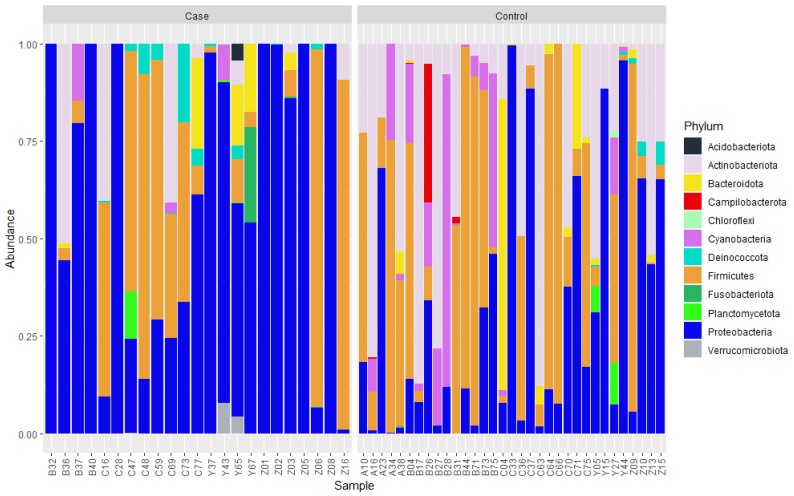
Relative abundance of bacterial phyla (*n* = 12) in the hemolymph of moribund mussels (Cases) and apparently normal healthy mussels (Controls) collected in 2017 and 2018 from six sites on the Clinch River, Virginia and Tennessee, USA, as determined by 16SrRNA gene sequence counts.

**Figure 4 microorganisms-09-01976-f004:**
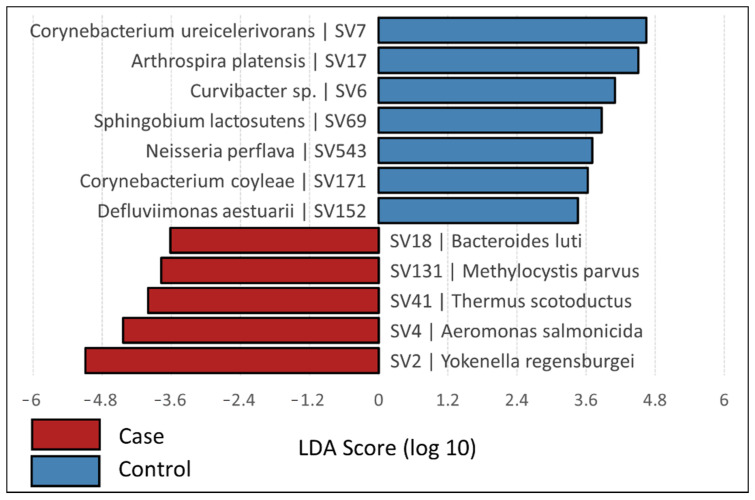
LEfSe analysis displaying differentially abundant SVs between cases and controls, based on a linear discriminant analysis with *p* < 0.05 for Kruskal–Wallis tests. Each SV is identified with the nearest BLAST result.

**Figure 5 microorganisms-09-01976-f005:**
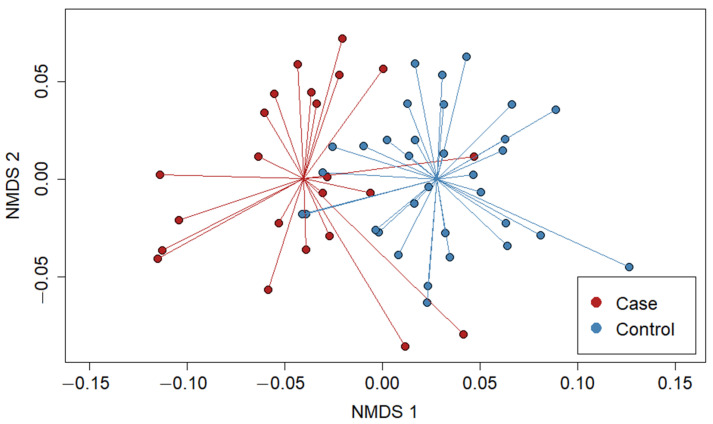
Non-metric multidimensional scaling (NMDS) of bacterial community composition of *A. pectorosa* at the SV level. Plot shows NMDS axes 1 and 2. Lines connecting points of each group (cases and controls) correspond to group centroids (average coordinates).

**Figure 6 microorganisms-09-01976-f006:**
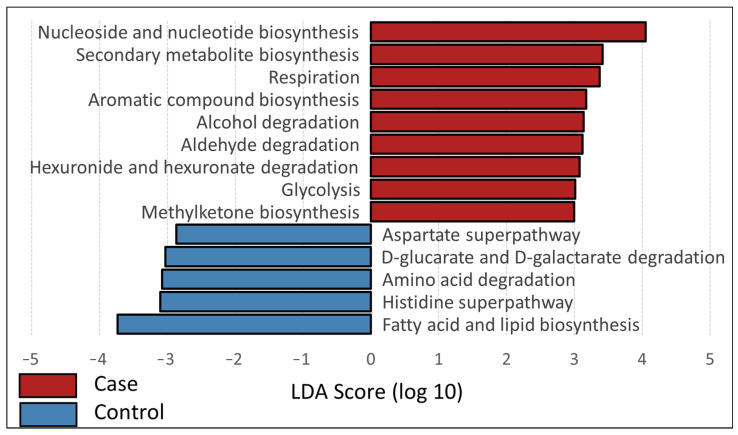
LEfSe analysis displaying differentially abundant second-level MetaCyc pathways between cases and controls, based on a linear discriminant analysis with *p* < 0.05 for Kruskal–Wallis tests.

## Data Availability

Raw sequences are deposited in the NCBI Sequence Reads Archive under the BioProject PRJNA756308.

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
