# Peer review of "Mussel Mass Mortality and the Microbiome: Evidence for Shifts in the Bacterial Microbiome of a Declining Freshwater Bivalve"

_microorganisms, 2021, doi:10.3390/microorganisms9091976_

Round 1
Reviewer 1 Report
The study constitutes a metabarcoding analysis of freshwater mussel hemolymph originating and potentially associated with a mortality event. The hypothesis of an association of the bacterial loads of each individual with clinical status and densovirus infection is really interesting and may provide valuable inferences. The manuscript is very well written and methodologically well organized. There are, however, some recommendations that follow and have be considered before publication. Introduction Since not all readers are familiar with the members of the family Unionidae, the authors should report if pheasant shells (Actinonaias pectorosa) are edible bivalves. This info is of high importance when associated with particular pathogenic bacteria that may be identified and could potentially be transferred to human. In the second paragraph of the Introduction the authors should add at least one reference regarding mass mortalities in Pinna nobilis throughout the Mediterranean Sea that have been attributed to Haplosporidium pinnae, Mycobacterium sp., or other potentially other microorganisms (Lattos et al. 2020, Pathogens, 9, 1002; doi:10.3390/pathogens9121002). Also, keeping in mind that Clinch densovirus constitutes a novel virus, the authors should add a short part of 1-2 sentences about this virus. 2.1 Were the specimens sacrificed or anaesthetized to obtain the hemolymph? 2.2 In 2.2 the authors describe apart from the DNA extraction, the PCR and the sequencing. Thus the title has to be modified. 2.5 How were animals classified as positive or negative for Clinch densovirus 1? Please provide some details 3.3 qPCR data should be corrected to "3.1 qPCR data", "3.1 Bacterial Community Composition" to 3.2 and "3.2 Inferred Functional Profiles" to 3.3 Discussion It would be very interesting if the researchers could provide a figure of the hemolymph collected from healthy mussels that appeared clear as well as of the cloudy appearance, if possible Also, it would be also interesting to provide a figure of a moribund individual The finding that Proteobacteria dominate bacterial communities in bivalves is a finding observed previously as well, in both alive individuals and ones suffering from mortalities (Lattos et al. 2020, Pathogens, 9, 1002; doi:10.3390/pathogens9121002). Any possible inference regarding this finding? One would expect cyanobacteria to be present majorly in moribund individuals, on the contrary with the observed findings. Maybe this finding should be further discussed. Finally, temperature, in terms of heat or cold and oxidative stress, is not discussed at all. The authors should add a small part, at least to exclude this possibility.
Author Response
Point 1: The manuscript by Richard and colleagues investigates the causes underlying the mass mortality events recorded in the phaesantshell A. pectorosa by analyzing the associated microbial communities. This is an interesting paper, which tries to shed some light on a complex and still poorly understood topic. Bivalve MMEs have been largely documented over the past few decades and those occurring in freshwater bivalves have obvious important implications on the conservation of these vulnerable and often endangered species, as well as on the whole river ecosystem. Nevertheless, the causes underlying such events are not entirely clear, as several different factors may concur. Shifts in the composition of microbial communities are one of the main candidates in explaining these events, so any study addressing this issue is a welcome addition.
Response 1: Many thanks for the positive comments and appraisal.
Point 2: Introduction - Page 2: I would suggest to indicate D. bugensis as “D. rostriformis bugensis”, as several readers may be familiar with the alternate scientific name of the species (both are currently accepted).
Response 2: This is indeed a valid point; many thanks. We have changed the reference to “D. rostriformis bugensis” per the reviewer’s suggestion (line 48).
Point 3: “In oysters and marine mussels”: replace with “In marine bivalves”. Several oysters are freshwater species, so this may lead to some confusion in the readers, and MMEs are often recorded in other types of bivalves, such as clams, scallops, etc., with Pinna nobilis being an additional possible recent example.
Response 3: The reviewer is indeed correct. Referencing “marine bivalves” more accurately reflects the scope of the relevant. We have replaced the language as suggested (Lines 56-57)
Point 4: Materials and methods - Is there any specific reason why hemolymph samples and not other tissues have been collected? Previous studies have demonstrated that different microbial communities are associated with different bivalve tissues, which may be also the case of pheasantshells. While the hemolymph most certainly represents a highly important target due to its involvement in immune response, other tissues may be equally (or even more) important to address MMEs in this species, such as mantle, gills and digestive gland.
Response 4: We thank the reviewer for raising this point. We chose hemolymph as our sampling target intentionally because it can be collected non-lethally, as validated by peer-reviewed publications and because it is not as exposed to the ambient environment as our tissue surfaces (i.e., similar to vertebrate blood). To clarify our reasoning, we have added details regarding our choice of hemolymph to the methods section (Lines 105-107).
Point 5: Replace “300 x 300 cycle chemistry” with “2 x 300 cycle chemistry”
Response 5: This is a great catch, thank you! We have replaced the text as suggested by the reviewer (Line 143).
Point 6: Page 5: please specify the BLAST e-value threshold used for this assessment.
Response 6: Thank you for making this suggestion. We have updated Table S2 with an additional column specifying the e-value of the top hit for each SV in the BLAST analysis.
Point 7: Results - Figure 2: please indicate statistically significant differences with an asterisk
Response 7: We thank the reviewer for catching this. We have added an asterisk to Figure 2a to indicate a statistically significant difference between the two groups.
Point 8: Discussion - I appreciate the efforts made by the authors to explain the results obtained in hemolymph, with an appropriate discussion about the near-sterility of hemolymph in healthy mussels vs its apparent invasion by bacteria in diseased and moribound mussels. With this respect, the authors could maybe go a bit forward, discussing that significant alterations of microbial communities could also involve mucosal barriers, which are the first line of defense in filter-feeding organisms, and whose role in bivalve immunity has been increasingly studied over the past few years. While alterations of hemolyph-associated bacteria may represent a sort of an “irreversible” final-stage infection, more subtle alterations of microbial communities associated with the gills, the mantle or the digestive gland may possibly indicate early signs of the events that will eventually lead to MMEs.
Response 8: We are grateful to the reviewer for this insightful suggestion. We had not specifically considered the relationship between microbes in hemolymph and those on mucosal barriers, but it’s a great point. We have therefore added a section discussing this possibility and its implications for mussel health (Lines 420-427).
Point 9: The discussion does not report other possible causes underlying MMEs: what about pollution, heavy metal accumulation, oxygen availability, etc.? Have these ever been investigated in the context of Clinch river MMEs? May any of these be considered as possible secondary causes of these events? While MMEs in marine bivalves have been clearly linked with shifts in the composition of microbial communities, other environmental conditions are often thought to trigger these events, allowing these shifts to take place, such as heat waves in lagoons, altered freshwater and nutrient inputs from estuaries, etc.
Response 9: This is an excellent point, and mirrors that of Reviewer 1 point 13. We are in the process of conducting a comprehensive historical review to identify co-occurring environmental factors associated with the Clinch River MME. Thus far, water quality and other abiotic factors do not appear to be playing primary roles, although we suspect that they are modulators of health and disease states (i.e., contributing factors). We therefore have added a section to the discussion describing how abiotic factors might interact with microbial factors during MMEs (lines 499-503).
Reviewer 2 Report
The manuscript by Richard and colleagues investigates the causes underlying the mass mortality events recorded in the phaesantshell A. pectorosa by analyzing the associated microbial communities. This is an interesting paper, which tries to shed some light on a complex and still poorly understood topic. Bivalve MMEs have been largely documented over the past few decades and those occurring in freshwater bivalves have obvious important implications on the conservation of these vulnerable and often endangered species, as well as on the whole river ecosystem. Nevertheless, the causes underlying such events are not entirely clear, as several different factors may concur. Shifts in the composition of microbial communities are one of the main candidates in explaining these events, so any study addressing this issue is a welcome addition.
Introduction
Page 2: I would suggest to indicate D. bugensis as “D. rostriformis bugensis”, as several readers may be familiar with the alternate scientific name of the species (both are currently accepted).
“In oysters and marine mussels”: replace with “In marine bivalves”. Several oysters are freshwater species, so this may lead to some confusion in the readers, and MMEs are often recorded in other types of bivalves, such as clams, scallops, etc., with Pinna nobilis being an additional possible recent example.
Materials and methods
Is there any specific reason why hemolymph samples and not other tissues have been collected? Previous studies have demonstrated that different microbial communities are associated with different bivalve tissues, which may be also the case of pheasantshells. While the hemolymph most certainly represents a highly important target due to its involvement in immune response, other tissues may be equally (or even more) important to address MMEs in this species, such as mantle, gills and digestive gland.
Replace “300 x 300 cycle chemistry” with “2 x 300 cycle chemistry”
Page 5: please specify the BLAST e-value threshold used for this assessment.
Results
Figure 2: please indicate statistically significant differences with an asterisk
Discussion
I appreciate the efforts made by the authors to explain the results obtained in hemolymph, with an appropriate discussion about the near-sterility of hemolymph in healthy mussels vs its apparent invasion by bacteria in diseased and moribound mussels. With this respect, the authors could maybe go a bit forward, discussing that significant alterations of microbial communities could also involve mucosal barriers, which are the first line of defense in filter-feeding organisms, and whose role in bivalve immunity has been increasingly studied over the past few years. While alterations of hemolyph-associated bacteria may represent a sort of an “irreversible” final-stage infection, more subtle alterations of microbial communities associated with the gills, the mantle or the digestive gland may possibly indicate early signs of the events that will eventually lead to MMEs.
The discussion does not report other possible causes underlying MMEs: what about pollution, heavy metal accumulation, oxygen availability, etc.? Have these ever been investigated in the context of Clinch river MMEs? May any of these be considered as possible secondary causes of these events? While MMEs in marine bivalves have been clearly linked with shifts in the composition of microbial communities, other environmental conditions are often thought to trigger these events, allowing these shifts to take place, such as heat waves in lagoons, altered freshwater and nutrient inputs from estuaries, etc.
Author Response
Point 1: The study constitutes a metabarcoding analysis of freshwater mussel hemolymph originating and potentially associated with a mortality event. The hypothesis of an association of the bacterial loads of each individual with clinical status and densovirus infection is really interesting and may provide valuable inferences. The manuscript is very well written and methodologically well organized. There are, however, some recommendations that follow and have be considered before publication.
Response 1: We thank the reviewer for the careful consideration of our manuscript and helpful suggestions.
Point 2: Introduction - Since not all readers are familiar with the members of the family Unionidae, the authors should report if pheasant shells (Actinonaias pectorosa) are edible bivalves. This info is of high importance when associated with particular pathogenic bacteria that may be identified and could potentially be transferred to human.
Response 2: We thank the reviewer for raising this important point. While North American Unionids were historically utilized as a food source by Native Americans, they are generally not consumed by humans due to their unpalatability, bioaccumulation of toxins, and regulations prohibiting their collection. We have therefore added a statement in the introduction clarifying this point (lines 49-50).
Point 3: In the second paragraph of the Introduction the authors should add at least one reference regarding mass mortalities in Pinna nobilis throughout the Mediterranean Sea that have been attributed to Haplosporidium pinnae, Mycobacterium sp., or other potentially other microorganisms (Lattos et al. 2020, Pathogens, 9, 1002; doi:10.3390/pathogens9121002).
Response 3: The reviewer makes an excellent point and is clearly knowledgeable about relevant recent literature from marine bivalves and mass mortality events. We have reviewed our list of cited literature for prominent marine bivalve cases and agree that recent literature regarding Pinna nobilis mass mortalities in the Mediterranean Sea is critical to include. We have therefore added a section to the introduction discussing the case of P. nobilis in the Mediterranean (lines 59-64).
Point 4: Also, keeping in mind that Clinch densovirus constitutes a novel virus, the authors should add a short part of 1-2 sentences about this virus.
Response 4: We are grateful to the reviewer for making this suggestion. We are engaged in ongoing research to learn more about Clinch densovirus 1, but knowledge gaps still remain. We have included additional details about the relationship between the virus and pheasantshell mortality in the introduction (line 84-86).
Point 5: 2.1 Were the specimens sacrificed or anaesthetized to obtain the hemolymph?
Response 5: This is an excellent point that we had not addressed in the manuscript, we thank the reviewer for raising it. Animals were not sacrificed as part of this study, and anesthesia is not used for sampling hemolymph from bivalves, which lack a central nervous system. We have included an explanation of these points along with additional details regarding our use of hemolymph for sampling as per Point 4 from reviewer 2 (Lines 105-107).
Point 6: 2.2 In 2.2 the authors describe apart from the DNA extraction, the PCR and the sequencing. Thus the title has to be modified.
Response 6: Good catch, thank you! We have changed the title of section 2.2 to “DNA Extraction, PCR, and Sequencing” (line 116)
Point 7: 2.5 How were animals classified as positive or negative for Clinch densovirus 1? Please provide some details
Response 7: Thank you for raising this point. Animals were classified based on the presence or absence of any NGS reads mapping to the genome of Clinch densovirus 1, described previously in Richard et al. (2020). We agree that it is important to describe this explicitly in the current manuscript, and have added an explanation to this effect (lines 195-196).
Point 8: 3.3 qPCR data should be corrected to "3.1 qPCR data", "3.1 Bacterial Community Composition" to 3.2 and "3.2 Inferred Functional Profiles" to 3.3
Response 8: Thank you for spotting this. We have corrected the numbering as corrected by the reviewer (Lines 258, 275, 361).
Point 9: Discussion - It would be very interesting if the researchers could provide a figure of the hemolymph collected from healthy mussels that appeared clear as well as of the cloudy appearance, if possible
Response 9: We agree completely with the reviewer! Unfortunately, we did not think to take photographs at the time of collection and processing, as we were unaware of the potential significance of the observation. We will make every effort to document this phenomenon during future sampling efforts.
Point 10: Also, it would be also interesting to provide a figure of a moribund individual Response 10: As above, we agree with the reviewer that a figure of a moribund individual would be an excellent addition. We have added an example of a moribund individual to the supplemental figures as Figure S9.
Point 11: The finding that Proteobacteria dominate bacterial communities in bivalves is a finding observed previously as well, in both alive individuals and ones suffering from mortalities (Lattos et al. 2020, Pathogens, 9, 1002; doi:10.3390/pathogens9121002). Any possible inference regarding this finding?
Response 11: We thank the reviewer for raising this important point. We had referenced proteobacteria dominance in stressed captive unionids in our discussion, but had not referenced similar observations in diseased marine bivalves. To address this point, we have added the references suggested by the reviewer to the discussion (lines 436-438).
Point 12: One would expect cyanobacteria to be present majorly in moribund individuals, on the contrary with the observed findings. Maybe this finding should be further discussed.
Response 12: The reviewer is indeed correct that the patterns exhibited by cyanobacteria in our study are contrary to our predictions. Interestingly, we have learned that several ongoing studies of hellbenders (a type of salamander) in rivers have shown the same association of cyanobacteria with healthy controls – but not moribund individuals. We have therefore described the observed pattern and speculated that, if substantiated, this type of association may be characteristic of riverine systems (lines 474-477).
Point 13: Finally, temperature, in terms of heat or cold and oxidative stress, is not discussed at all. The authors should add a small part, at least to exclude this possibility.
Response 13: The reviewer makes an excellent point, similar to that of Reviewer 2, Point 9 regarding additional abiotic factors that warrant discussion as potentially related to observed morbidity and mortality in pheasantshell. Based on the reviewer's excellent suggestions, we have added a section to the discussion to this address the potential involvement of environmental effects (lines 499-503).
Round 2
Reviewer 1 Report
The requested modifications are conducted and therefore the work is suitable for publication.